# COMPUTATIONAL-UNIDENTIFIABILITY IN REPRESENTATION FOR FAIR DOWNSTREAM TASKS

## ABSTRACT

Deep representation learning methods are highlighted as they outperform classical algorithms in various downstream tasks, such as classification, clustering, generative models, etc. Due to their success and impact on the real world, fairness concern is rising with noticeable attention. However, the focus of the fairness problem was limited to a certain downstream task, mostly classification. We claim that the fairness problems to various downstream tasks originated from the input feature space, *i.e.,* the learned representation space. While several studies explored fair representation for the classification task, the fair representation learning method for unsupervised learning is not actively discussed yet. To fill this gap, we define a new notion of fairness, *computational-unidentifiability*, which suggests the fairness of the representation as the distributional independence of the sensitive groups. We demonstrate motivating problems that achieving computationally-unidentifiable representation is critical for fair downstream tasks. Moreover, we propose a novel fairness metric, Fair Fréchet distance (FFD), to quantify the *computational-unidentifiability* and address the limitation of a well-known fairness metric for unsupervised learning, *i.e.,* balance. The proposed metric is efficient in computation and preserves theoretical properties. We empirically validate the effectiveness of the *computationally-unidentifiable* representations in various downstream tasks.

## 1 INTRODUCTION

Thanks to the outstanding performance and development of deep learning, it has been widely applied to various domains, including natural language processing (NLP) (Devlin et al., 2018), computer vision (Karras et al., 2019), and generative models (Goodfellow et al., 2014). On the other hand, the reliability and fairness concerns (Lee & Floridi, 2020; Angwin et al., 2016; Dastin, 2018) advanced due to their impact on the real world applications. Such fairness concerns include credit limit estimation (Vigdor, 2019), job application filtering (Dastin, 2018), or crime prevention (Dressel & Farid, 2018), etc. Accordingly, algorithmic fairness is getting growing attention to prevent biased predictions.

Following the mainstream fairness literature, we here focus on group fairness (Dua & Graff, 2019; Zafar et al., 2015; Hardt et al., 2016), which suggests the equality of certain statistical measures (*e.g.,* true positive rate, positive prediction) between subgroups with different protected attribute (*e.g.,* gender, race, religion, *etc*). It has been widely studied to mitigate fairness violations in downstream tasks. Numerous studies (Hardt et al., 2016; Choi et al., 2020; Pleiss et al., 2017; Madras et al., 2018) explore how to attain group fairness in classification tasks. The primary objective of this family of works is to obtain the prediction independence of a protected property. Hardt et al. (2016) suggest equal opportunity, which requires the same true positive rates for the subgroup. Calibration among the subgroups (Kleinberg et al., 2016) is to match the predicted probability and actual distribution of favorable class. Moreover, some works (Kim et al., 2020; Jang et al., 2021) study efficient multi-constraint optimization to satisfy multiple fairness notions.

However, most of the works mainly focus on the supervised setting. Even though deep learning has significant success in various unsupervised learning tasks, such as clustering (Xie et al., 2016; Guo et al., 2017), generative model (Karras et al., 2019; Radford et al., 2019), and NLP (Hadifar et al., 2019), the fairness of unsupervised learning is relatively not actively studied (Buet-Golfouse & Utyagulov, 2022), and how to quantify the fairness of unsupervised learning methods has not been

well established yet. A widely used metric for fair clustering is called *balance* (Chierichetti et al., 2017), which is analogous to demographic parity (Barocas & Selbst, 2016) in classification. However, the balance has some limitations since it quantifies fairness by computing the ratio of samples in different protected groups within a cluster. For instance, even in the ideal balance (the ratio of the samples from different groups matches the group truth), the sensitive groups can distribute samples separately within clusters. In this case, it is easy to determine which sensitive group the sample belongs to so that it might lead to a biased decision in downstream tasks. Especially in generative models, *e.g.,* VAE (Kingma & Welling, 2013), the generated samples can be imbalanced if the latent space is dependent on the sensitive attributes. This can cause a critical problem as generative models are widely applied to mitigate the imbalance of datasets (Guo et al., 2019; Fajardo et al., 2021; Mirza et al., 2021).

Instead, we propose a novel approach, *computational-unidentifiability*, as a fairness notion in unsupervised learning. Analogous to the fact that biased data is responsible for the biased decision-making (Buolamwini & Gebru, 2018; Mehrabi et al., 2021), we here claim that the learned representation itself plays a critical role in fair downstream tasks utilizing DNN. Even though deep representation has been appreciated for its superb performance (Eldan & Shamir, 2016; Kozma et al., 2018), the fairness concerns in the space have been overlooked. Thus, we explore the fairness in representation space that could bridge DNN and the downstream tasks with fairness concerns. We validate our claim on downstream tasks by comparing the performance and fairness of two distributions: fair and unfair representation.

To measure fairness in representation space, we propose a novel metric called FFD (Fair Fréchet distance) inspired by Fréchet distance (Dowson & Landau, 1982) to efficiently quantify fairness in representation space by measuring distributional independence of the sensitive groups with computational identifiability (Hébert-Johnson et al.; Lahoti et al., 2020). Unlike the balance, we not only consider *statistical independence* but also *distributional independence* between the sensitive groups. This can be a good reference for future work to evaluate the fairness or distributional independence in the representation space of certain attributes of interest. Moreover, we propose a deep fair clustering framework to learn a fair representation that achieves comparable performance with other clustering methods while ensuring fairness. The contributions in the paper can be summarized as follows:

1. We study the motivating problem of why fair representation is important to achieve fair downstream tasks.

2. We propose a novel metric that quantifies fairness in representation space. We provide rigorous analysis of the theoretical property and complexity of our fairness metric.

3. We propose a framework for fair representation learning for downstream tasks.

4. We validate our method on various benchmark datasets comparing with state-of-the-art fair methods in the literature.

## 2 RELATED WORKS

### GROUP FAIRNESS

As a class of definitions, group fairness measures the disparity of predicted outcomes among the subgroups with certain sensitive attributes. A number of works introduce fair notions to mitigate the bias and ensure the independence of the performance measures between the subgroups to achieve group fairness. Demographic parity (Barocas & Selbst, 2016) suggests that positive prediction should be equalized and independent of the sensitive attribute. Equal opportunity (Hardt et al., 2016) states that true positive rates should match. Likewise, Predictive equality (Chouldechova, 2017) states the equality of false positive rates. Group-wise calibration (Kleinberg et al., 2016; Pleiss et al., 2017) proposed to match the probability estimate with the actual ratio of positive distribution within the group. In an unsupervised setting, balance (Chierichetti et al., 2018) is introduced to have an equal number of samples from different protected groups within a cluster as fair clustering. However, the balance only considers statistical parity, which limits the utility as a metric since perfect balance (*i.e.,* 1) does not guarantee fairness (as the base rate differs). Moreover, none of the works explore the fairness of the representation itself.

FAIR SUPERVISED LEARNING

To assure group fairness, recent works in supervised learning reside in one of the three approaches: 1) pre-processing; 2) in-processing; 3) post-processing. Pre-processing method (Chen et al., 2018) suggests improving the skewed sample size problem. Adversarial learning methods (Madras et al., 2018; Zhao et al., 2019) are proposed to learn a fair representation that is independent of sensitive attributes. Post-processing the output of the biased model (Hardt et al., 2016; Jang et al., 2021) with multiple fairness objectives are introduced as they are more efficient than training a model from scratch. However, the aforementioned approaches minimize group fairness constraints specified for the classification task.

FAIR UNSUPERVISED LEARNING

Fairness in the unsupervised setting has recently got attention (Buet-Golfouse & Utyagulov, 2022; Ghadiri et al., 2021). A pioneering work (Chierichetti et al., 2018) in fair clustering method proposed fairlet decomposition to pre-process data followed by classic clustering methods to address disparate impact. Scalable fair clustering algorithm (Backurs et al., 2019) is the following work of fairlet decomposition by improving the efficiency with approximation. Variational framework (Ziko et al., 2019) is introduced to satisfy KL fairness objective. Wang & Davidson (2019) propose a new concept called fairoid that enforces the centroids of each sensitive group in feature space to have an equal distance to each cluster centroid. Adversarial objective (Li et al., 2020) is employed to learn a representation that is statistically independent *w.r.t.* sensitive attribute while clustering-favorable utilizing individual clustering modules. However, to our best knowledge, previous works mostly focused on the predicted outcome to be independent of the sensitive attribute, *i.e.,* statistical parity. We here study the independence of sensitive attributes in the learned representation in unsupervised learning.

## 3 MOTIVATING PROBLEMS

In this work, we define a novel fairness notion called *computationally unidentifiability* that is more extensive than the existing task-specific notions. Inspired by fair classification works (Hébert-Johnson et al.; Lahoti et al., 2020), we define computational-identifiability as the maximum possible ability for an external classifier to distinguish which sensitive group the data belongs to. Two distributions are computationally unidentifiable if and only if they are identical, *i.e.,* no external classifier can distinguish which sensitive group the sample is drawn from. We demonstrate motivating problems showing how such distributional independence affects fairness in downstream tasks.

### 3.1 CLASSIFICATION AND CLUSTERING

Consider data distribution with binary sensitive attribute, $A = \{0, 1\}$, and binary label, $Y = \{0, 1\}$. In Fig. 1, we illustrate synthetic data distributions similar to previous works (Zafar et al., 2015; Kim et al., 2020) with two scenarios that both satisfy the perfect balance, *i.e.,* base rate for each protected group is identical. The perfect balance can also be referred to as *statistical independence*. We denote $X_{ya}$ as a set of instances with $y \in Y$ and $a \in A$. The detail of the synthetic data sampling process is in the appendix. When comparing two distributions in Fig. 1b and Fig. 1a, the distribution in Fig. 1b explicitly exposes which sensitive group a sample belongs to, *i.e.,* computationally identifiable (CI). This has a potential risk of discrimination in downstream tasks. Specifically, it is unstable for maintaining good clustering performance since such data representation can be clustered by the sensitive group structure whether than the expected intrinsic features (Lee et al., 2021). In contrast, in Fig. 1a, the representation satisfies not only statistical independence but also *distributional independence w.r.t.* sensitive attribute, *i.e.,* computationally unidentifiable (CU). Therefore, models cannot easily identify which group a sample belongs to and thus cannot discriminate against groups in the downstream tasks.

To validate our claim, we evaluate two representations with classification and clustering, which are the most popular tasks in supervised and unsupervised learning. For the classification task, we test the logistic regression. To measure fairness in classification, we adopt demographic parity (Barocas & Selbst, 2016), $DP = |P(\hat{Y} = 1|A = 0) - P(\hat{Y} = 1|A = 1)|$, and equalized odds (Hardt et al.,

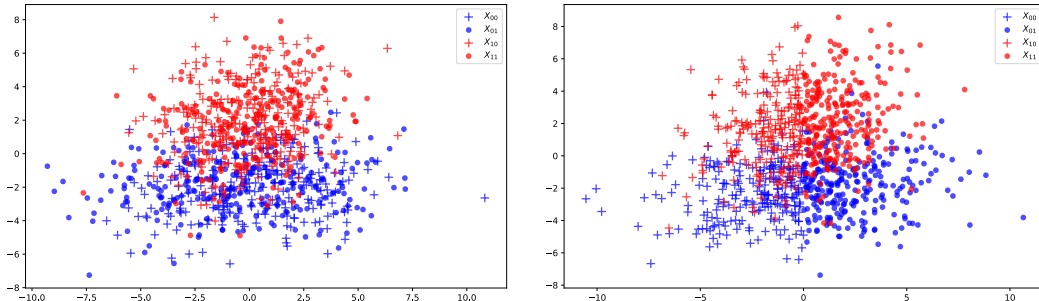

(a) Computationally unidentifiable (fair) representation    (b) Computationally identifiable (unfair) representation

Figure 1: Illustration of two synthetic data of computationally unidentifiable (fair) and identifiable (unfair) distribution with a binary sensitive group and class. Different color (*resp.* shape) indicates different class $y$ (*resp.* sensitive group $a$). We denote $X_{ya}$ as a set of instances with $y, a \in \{0, 1\}$.

Table 1: Evaluation of downstream tasks on different distributions. Computational-Unidentifiable (CU) distribution achieves significantly fairer results with similar performance in both supervised and unsupervised learning tasks, while Computationally-Identifiable (CI) distribution has a huge detriment of fairness. FFD is the proposed metric to measure the fairness of a representation.

|  | Data | | $k$-means++ (Unsup Learning) | | | Logistic Regression (Sup Learning) | | | |
|---|---|---|---|---|---|---|---|---|---|
|  | $FFD^2$ | Balance | ACC | $\Delta$ACC | Balance | ACC | $\Delta$ACC | DP | EOD |
| CU | 6.876 | 0.931 | 0.846 | 0.002 | 0.851 | 0.865 | 0.005 | 0.029 | 0.031 |
| CI | 8.385 | 0.908 | 0.861 | 0.091 | 0.116 | 0.855 | 0.076 | 0.636 | 0.717 |

2016), EOD $= \sum_y |P(\hat{Y} = y|A = 0, Y = y) - P(\hat{Y} = y|A = 1, Y = y)|$, where $\hat{Y}$ is predicted label. Both DP and EOD are the lower, the better.

For fair clustering, we measure the balance by following the previous works (Xie et al., 2016; Li et al., 2020; Bera et al., 2019), which is to satisfy $\mathbb{E}_{X \sim \mathcal{D}}[A = a|C(X) = k] = \mathbb{E}_{\mathcal{X} \sim \mathcal{D}}[A = a]$, where $C(X) = k$ indicates that the data $X$ is clustered to the $k$-th cluster by model $C$. Achieving the balanced clustering satisfies the statistical independence *w.r.t.* $A$, and balance $= 1$ is a perfect balance. However, we claim that statistical independence cannot fully examine fair clustering. To address the limitation of the previous fair unsupervised learning metric, we propose a novel fairness metric for representation called Fair Fréchet Distance (FFD), which will be discussed in the following section.

Table 1 summarizes the evaluation of downstream tasks on the two distributions. Even though both CU and CI data are sampled from perfectly balanced distributions, fairness violations from CI are significantly worse than that of CU on both tasks. It is interesting to note that fairness is sensitive to distributional independence; however, performance is not affected. This validates that fair representation itself has a substantial impact on fairness in downstream tasks while preserving utility. Moreover, FFD is a good proxy to measure computational identifiability since smaller FFD indicates harder to identify sensitive information from the representation.

## 4    FAIR FRÉCHET DISTANCE

To quantify the proposed fairness notion in terms of computational identifiability, in this subsection we introduce a novel metric named Fair Fréchet Distance (FFD) to measure the distance between distributions from different sensitive groups.

Consider two sets of samples $U \in \mathbb{R}^{d \times n_0}$ and $V \in \mathbb{R}^{d \times n_1}$. Suppose the samples in $U$ and $V$ are drawn from multivariate Gaussian distributions, respectively. Define a centering matrix $H_n \in \mathbb{R}^{n \times n}$ as $H_n = I - \frac{1}{n}\mathbf{1}_n\mathbf{1}_n^\top$, where $\mathbf{1}_n \in \mathbb{R}^n$ is a vector with all elements being 1; and $I$ is the identity matrix. We first introduce two metrics in Definition 4.1 and 4.2 that measure the distance between two distributions $U$ and $V$.

**Definition 4.1.** Fréchet distance (FD) (Dowson & Landau, 1982) between $U$ and $V$ is defined as:

$$FD^2(U,V) = \|\mu_U - \mu_V\|_2^2 + \text{Tr}\left(\Sigma_U + \Sigma_V - 2(\Sigma_U^{\frac{1}{2}}\Sigma_V\Sigma_U^{\frac{1}{2}})^{\frac{1}{2}}\right),$$

where $\mu_U, \mu_V$ and $\Sigma_U, \Sigma_V$ are the means and covariance matrices of $U$ and $V$, respectively.

**Definition 4.2.** We define the Fair Fréchet Distance within Cluster (FFDC) between $U$ and $V$ as follows:

$$FFDC^2(U,V) = \left\|\frac{U\mathbf{1}_{n_0}}{n_0} - \frac{V\mathbf{1}_{n_1}}{n_1}\right\|_2^2 + \left(\frac{\|UH_{n_0}\|_F}{\sqrt{n_0-1}} - \frac{\|VH_{n_1}\|_F}{\sqrt{n_1-1}}\right)^2 + \frac{\text{Tr}(UU^\top) + \text{Tr}(VV^\top)}{\sqrt{n_0-1}\sqrt{n_1-1}},$$

when $n_0, n_1 > 1$, else $FFDC^2(U,V) = \infty$.

Next, we define Fair Fréchet Distance (FFD) in Definition 4.3. For simplicity, we define FFD for the case with binary sensitive feature $a \in \{0,1\}$. We will introduce how to extend such a measure to the case with multi-valued sensitive features at the end of this subsection.

For a clustering assignment of $m$ samples into $c$ clusters as $\{X_1, X_2, \ldots, X_k\}$, where $X_k \in \mathbb{R}^{d \times n_k}, k = 1, 2, \ldots c$, contains the $n_k$ samples in the $k$-th cluster that sums to $\sum_{k=1}^{c} n_k = m$. Within each cluster $X_k$, define $U_k \in \mathbb{R}^{d \times n_k}$ and $V_k \in \mathbb{R}^{d \times n_k}$ as follows:

$$\mathbf{u}_k^i = \begin{cases} \mathbf{x}_k^i, & \text{if } a_k^i = 0 \\ \mathbf{0}, & \text{else} \end{cases}, \quad \mathbf{v}_k^i = \begin{cases} \mathbf{x}_k^i, & \text{if } a_k^i = 1 \\ \mathbf{0}, & \text{else} \end{cases} \tag{1}$$

where $\mathbf{x}_k^i$ is the $i$-th sample in $X_k$; $a_k^i \in \{0,1\}$ is the sensitive feature of the $i$-th sample in $X_k$; and $\mathbf{0}$ is the zero vector. Thus we have $U_k + V_k = X_k, k = 1, 2, \ldots, c$.

**Definition 4.3.** With the definition of $U_k|_{k=1}^c$ and $V_k|_{k=1}^c$ in equation 1, we define FFD for the $m$ samples with the clustering assignment $\{X_1, X_2, \ldots, X_k\}$ as:

$$FFD(\{X_1, X_2, \ldots, X_k\}) = \max_k FFDC(U_k, V_k).$$

**Theorem 4.4.** *With the definition of $U_k|_{k=1}^c$ and $V_k|_{k=1}^c$ in equation 1, the following inequality holds:*

$$FFD^2(\{X_1, X_2, \ldots, X_k\}) - \max_k \frac{1}{n_k - 1}\text{Tr}(X_k X_k^\top)$$
$$\leq \max_k FD^2(U_k, V_k) \leq FFD^2(\{X_1, X_2, \ldots, X_k\}).$$

Proof of Theorem 4.4 is in the appendix. In the case of the multi-valued sensitive feature, we can extend the definition of FFDC in Definition 4.2 with the max FFDC value among all pairs of sensitive groups in a cluster, and thus extend the definition of FFD in Definition 4.3. We can easily verify that Theorem 4.4 still holds in the case with the multi-valued sensitive feature.

## 4.1 Insight from Theorem 4.4

Consider a clustering problem that partitions the $m$ samples into $c$ clusters, where each data sample is formulated as a $d$-dimensional vector. The FFD metric we proposed in Definition 4.3 is efficient in computation, which requires linear time *w.r.t.* the number of features and number of samples. The calculation of FFD in Definition 4.3 has a time complexity of $O(ndc)$ (since we only need to calculate the trace of matrix $UU^\top$ and $VV^\top$ in Definition 4.2, it requires linear instead of quadratic time *w.r.t.* $d$). In contrast, traditional FD metric in Definition 4.1 has a cubic time complexity *w.r.t.* number of features. The time complexity for calculating $FD$ is $O(c(nd^2 + d^3))$ (since it requires the computation of exact covariance matrices $\Sigma_U$ and $\Sigma_V$ in Definition 4.1 and the corresponding square root).

Theorem 4.4 indicates that the FD metric is upper bounded by our proposed FFD metric, thus minimizing FFD indicates the minimization of the upper bound of FD. Further, the gap between the FD and our FFD metric is bounded by $\max_k \frac{1}{n_k-1}\text{Tr}(X_k X_k^\top)$. Note that FFD is minimized if and only if the two following conditions are met in each cluster $X_k, k = 1, 2, \ldots c$:

$$U_k\mathbf{1}_{n_k} = V_k\mathbf{1}_{n_k}, \quad \|U_k H_{n_k}\|_F^2 = \|V_k H_{n_k}\|_F^2,$$

in which case we have $\text{FD}^2 = 0$. Thus FD value is minimized if and only if our proposed FFD metric is minimized.

## 5 FAIR CLUSTERING FRAMEWORK

In this section, we present our deep fair clustering framework and its objective functions. For simplicity, we consider sensitive attribute as binary feature. However, this can be easily extended to multiple sensitive attribute problems. Consider the $c$-clustering problem, given the i.i.d. sampled $m$ data samples $\mathcal{X} \in \mathbb{R}^{d \times m}$, where each sample is represented by a $d$-dimensional vector. Encoder $E$ learns a representation $\mathcal{Z} \in \mathbb{R}^{l \times m}$ and a clustering module $C$ takes $\mathcal{Z}$ as an input and outputs probability $P \in \mathbb{R}^{c \times m}$ of the predicted cluster as a soft label. The goal of $E$ and $C$ is to achieve computationally unidentifiable fairness and high clustering performance. Given a matrix $\mathcal{X} \in \mathbb{R}^{d \times m}$ with $m$ samples, we denote the $i$-th data point from $\mathcal{X}$ as its bold lower case letter with index in the superscript, *e.g.*, $\mathbf{x}^i$, and the $k$-th entry of a vector as a lower case letter *e.g.*, $x_k^i$.

### 5.1 CLUSTERING LOSS

Inspired by the previous works (Xie et al., 2016; Li et al., 2020), we employ clustering loss to learn the representation that is concentrated in the cluster centroids. Clustering module $C$ assigns probability that a sample $\mathbf{z}^i = E(\mathbf{x}^i)$ belongs to each cluster $k'$ by comparing with trainable cluster centroids $\mathbf{c}_k$ on Student t-distribution as:

$$p_k^i = \frac{(1 + \|\mathbf{z}^i - \mathbf{c}_k\|^2/\alpha)^{-\frac{\alpha+1}{2}}}{\sum_{k'}(1 + \|\mathbf{z}^i - \mathbf{c}_{k'}\|^2/\alpha)^{-\frac{\alpha+1}{2}}}, \tag{2}$$

where $p_k^i$ indicates the probability that $\mathbf{x}^i$ belongs to $k$-th cluster and $\alpha$ is the degree of freedom of Student t-distribution. Then, assign the target cluster $q_k^i$ by sharpening the soft assignment $p_k^i$ within a sensitive group $a$ as

$$q_k^i = \frac{(p_k^i)^2/\sum_{\mathbf{x}^j \in \mathcal{X}_a} p_k^j}{\sum_{k'}\left((p_{k'}^j)^2/\sum_{\mathbf{x}^j \in \mathcal{X}_a} p_{k'}^j\right)}, \tag{3}$$

which reinforce the confidence of the predicted cluster and prevent large clusters as a regularizer. We set empirical clustering loss $\hat{\mathcal{L}}_{cls}$ as KL divergence between $p_k$ and $q_k$ as

$$\hat{\mathcal{L}}_{cls} = KL(P\|Q) = \sum_{\mathbf{x} \in \mathcal{X}} \sum_k p_k \log \frac{p_k}{q_k}. \tag{4}$$

### 5.2 FAIRNESS LOSS

Our goal is to further improve fairness in the clustering task that sensitive group is not identifiable by the samples in a cluster. Recent work proposed to use fairoid (fair-centroid) (Wang & Davidson, 2019) that the centroid of each sensitive group should have equal distance to all cluster centroids. We claim that fairoid cannot guarantee fair representation since equidistance centroids can be perfectly separated by the cluster centroids.

To achieve computational-unidentifiability, we employ variational autoencoder (VAE) structure (Kingma & Welling, 2013) for the encoder to leverage the reparameterization trick. Then we can formulate the latent feature of an instance $\mathbf{x}^i$ as $\mathbf{z}^i = E(\mathbf{x}^i) = \boldsymbol{\mu}^i + \epsilon\boldsymbol{\sigma}$, where $\epsilon \sim \mathcal{N}(0, I)$, where $\boldsymbol{\mu}$ and $\boldsymbol{\sigma}$ are the mean and variance respectively. To enforce the learned representation independent of the sensitive attribute, we minimize the distance between distributions from a different protected group within a cluster, *i.e.*, $KL(p_{(a,k)}\|p_{(a',k)})$, where $p_{(a,k)}$ is a probability distribution of the samples in a sensitive group $a$ with predicted cluster $k$. Assume the distribution follows the Gaussian distribution as $p_{(a,k)} = \mathcal{N}(\boldsymbol{\mu}_{(a,k)}, Diag(\boldsymbol{\sigma}_{(a,k)}))$. Then our fairness objective to minimize KL divergence can be written as:

$$\mathcal{L}_{fair} = -\frac{1}{2}\left(2\log\left(\frac{\boldsymbol{\sigma}_{(a,k)}}{\boldsymbol{\sigma}_{(a',k)}}\right) - \frac{\boldsymbol{\sigma}_{(a,k)}^2 + (\boldsymbol{\mu}_{(a,k)} - \boldsymbol{\mu}_{(a',k)})^2}{\boldsymbol{\sigma}_{(a',k)}^2} + 1\right). \tag{5}$$

For the empirical loss $\hat{\mathcal{L}}_{fair}$, we use $\hat{\boldsymbol{\mu}}_{(a,k)} = \frac{1}{|\mathcal{X}_{a,k}|}\sum_{i \in \mathcal{X}_{a,k}} \boldsymbol{\mu}_i$ and $\hat{\boldsymbol{\sigma}}_{(a,k)} = \frac{1}{|\mathcal{X}_{a,k}|}\sum_{i \in \mathcal{X}_{a,k}} \boldsymbol{\sigma}^i$ as the empirical mean and variance where $\mathcal{X}_{a,k}$ is denoted as a set of instances predicted as cluster $k$ in group $a$, since we assume all samples are i.i.d.

To sum up, our final objective is to minimize the loss as follows:

$$\min_{E,C} \hat{\mathcal{L}}_{cls} + \hat{\mathcal{L}}_{fair}. \tag{6}$$

# 6 EXPERIMENTS

In this section, we compare fairness and the performance of the proposed method with the state-of-the-art methods.

## 6.1 EXPERIMENTAL SETUP

**Benchmark Dataset**. We use two image datasets and two tabular datasets to evaluate the methods. MNIST-USPS dataset consists of 60,000 MNIST [1], and 7,291 USPS[2] hand written gray scale digits. We consider the source of the image *i.e.,* MNIST, USPS as a sensitive attribute with $c = 10$ clustering problem. MTFL (Zhang et al., 2014) consists of 12,995 facial images and its landmark information. It also provides information such as gender and wearing glasses. By following (Li et al., 2020), we use wearing glasses or not as a sensitive attribute and $c = 2$ clustering problem with desired clustering attribute is gender.

We pre-process the image dataset by normalizing the pixel value. The normalization parameters are mean $= 0.1307$, std $= 0.3081$ for MNIST-USPS, and mean$= (0.3527, 0.3902, 0.4697)$, and std$= (1, 1, 1)$ for MTFL respectively.

**Comparing Methods**. To evaluate our method, we compare with the following related methods in the experiments. ScFC (Backurs et al., 2019) is non-deep fair clustering method that approximates fairlet decomposition algorithm in a linear run time. ALG (Bera et al., 2019) is non-deep fair clustering method that is based on k-median approach. DFC (Li et al., 2020) is a deep fair clustering method to learn fair and clustering-favorable representation by adversarial loss and cluster modules with an individual group. VFC (Ziko et al., 2019) is a variational framework for fair clustering with KL fairness as clustering objective.

As a baseline and reference, we use $k$-means++ and perfect clustering. We use the same backbone structure for deep fair clustering methods for the fair evaluation. For USPS-MNIST, we pretrain the encoder to reconstruct the original image as VAE following DFC (Li et al., 2020). For MTFL, we adopt ResNet50 (He et al., 2016) pretrained with ImageNet for the encoder. We used Adam optimizer (Kingma & Ba, 2014) with learning rate as $10^{-5}$. We implement all experiments on Nvidia Quadro RTX 6000 and Intel i9-9960X with 128GB RAM.

**Evaluation Metric**. For the evaluation, we measure performance with accuracy and NMI (Strehl & Ghosh, 2002), and fairness with accuracy difference between sensitive groups, balance, and FFD. The four metrics can be computed as:

$$\text{Accuracy} = \frac{\sum_{\mathbf{x}^i \sim \mathcal{X}} \mathbb{1}[\arg\max_k p_k^i = y^i]}{n}, \quad \text{NMI} = \frac{\sum_{k,j} n_{kj}^+ \log\left(\frac{nn_{kj}^+}{n_k n_j^+}\right)}{\sqrt{\left(\sum_k n_k \log\frac{n_k}{n}\right)\left(\sum_j n_j^+ \log\frac{n_j^+}{n}\right)}},$$

$$\text{Balance} = \min_k \left(\min\left(\frac{n_{uk}}{n_{vk}}, \frac{n_{uk}}{n_{vk}}\right)\right),$$

$$\text{FFD}^2 = \max_k \left(\left\|\frac{U_k \mathbf{1}_{n_k}}{n_k} - \frac{V_k \mathbf{1}_{n_k}}{n_k}\right\|^2 + \left(\frac{\|U_k H_{n_k}\|_F}{\sqrt{n_k - 1}} - \frac{\|V_k H_{n_k}\|_F}{\sqrt{n_k - 1}}\right)^2 + \frac{\text{Tr}(U_k U_k^\top) + \text{Tr}(V_k V_k^\top)}{n_k - 1}\right).$$

We denote $n, n_k, n_{uk}, n_j^+$, and $n_{kj}^+$ as total number of samples, number of samples predicted as cluster $k$, cluster $k$ with group $u$, has ground truth label $j$, and samples intersected with $k$ and $j$. Also, $y^i$ indicates the true label of $\mathbf{x}^i$, which is matched to the clusters by the linear sum assignment problem to find the best pair between the predicted cluster and the true label for calculating accuracy. The lower bound of Fréchet Distance (FD) can be calculated by simply omitting the last term in the above equation.

---

[1] http://yann.lecun.com/exdb/mnist/
[2] https://www.kaggle.com/bistaumanga/usps-dataset

|  | MNIST-USPS | | | | MTFL | | | |
|---|---|---|---|---|---|---|---|---|
|  | Acc | NMI | Balance | FFD$^2$ | Acc | NMI | Balance | FFD$^2$ |
| Perfect | 1.0 | 1.0 | 0.12 | 14.90 | 1.0 | 1.0 | 0.145 | 7.57 |
| $k$-means++ | 0.593 | 0.503 | 0.054 | 16.24 | 0.727 | 0.228 | 0.076 | 7.57 |
| ScFC (Backurs et al., 2019) | 0.353 | 0.258 | 0.072 | 13.63 | 0.731 | 0.182 | 0.080 | 8.72 |
| ALG (Bera et al., 2019) | 0.581 | 0.487 | 0.091 | 15.80 | 0.747 | 0.227 | 0.124 | 8.80 |
| VFC (Ziko et al., 2019) | 0.580 | 0.522 | 0.115 | 14.67 | 0.756 | 0.227 | 0.140 | 8.79 |
| DFC (Li et al., 2020) | 0.824 | 0.828 | 0.053 | 14.13 | 0.731 | 0.163 | 0.137 | 48.02 |
| Ours (only $\mathcal{L}_{cls}$) | 0.768 | 0.762 | 0.058 | 2.04 | 0.762 | 0.211 | 0.081 | 46.30 |
| Ours | 0.831 | 0.837 | 0.091 | 1.82 | 0.728 | 0.154 | 0.113 | 43.58 |

Table 2: Evaluation of clustering methods on two datasets: MNIST-USPS and MTFL. For accuracy and NMI, it is higher the better. Balance is better if it is closer to perfect clustering *i.e.,* original data statistic. FFD$^2$ measures distributional independence of sensitive attribute, *i.e.,* the lower, the better. FFD is measured in the learned representation (*resp.* input space) space for the deep (*resp.* non-deep) models. FFD measurement in the input space is underlined.

## 6.2 QUANTITATIVE EVALUATION

In Table 2, we report the quantitative evaluation of two image datasets. For accuracy and NMI, the higher, the better, and balance is better if it is close to that of perfect clustering. For FFD, it is lower the better. To calculate FFD, we set all comparing deep models (DFC and ours) to have the same dimension in the representation space. In addition, we preprocessed the latent features from each model by normalizing the maximum magnitude to 1 for a fair comparison. For non-deep models, we measure FFD in the original input space, and the values are underlined. Note that we do not directly compare FFD from deep and non-deep models since they are calculated in different spaces.

In the table, we observe some non-deep fairness methods achieve lower accuracy than classical $k$-means++, which is sacrificed to have better balance. With the proposed method, we achieve comparable or better results on both accuracy and balance compared with the baselines. Moreover, we could achieve a significantly lower FFD than the other deep fair method, DFC (Li et al., 2020). As an ablation study, we evaluate our framework with the same structure without the fair loss term. We empirically found that integrating $\mathcal{L}_{fair}$ in training sometimes favorably contributes to not only fairness but also performance.

It is interesting to note that ScFC (Backurs et al., 2019) got lower FFD than the perfect clustering in MNIST-USPS. Thus FFD can be also a good measure to reveal how biased the dataset itself is against some demographic groups, *e.g.,* imbalanced data or under-representation analysis.

## 6.3 QUALITATIVE ANALYSIS

In this subsection, we qualitatively evaluate fairness of the learned representation proposed in the paper comparing with other deep fair clustering methods. Fig. 2 illustrates t-SNE (Van der Maaten & Hinton, 2008) visualization of the original data, the learned representation of our model, and DFC on MNIST-USPS dataset. The colors in top and bottom rows indicate different ground truth labels and sensitive attributes, respectively. The first two columns show the progress of our model in the training process. The last column in the figure is the visualization of DFC after it converges. At the starting phase, as in Fig. 2a, we observe that representation is clustered based on the sensitive attribute. This shows that for the pretraining of the encoder or some downstream tasks, sensitive information takes an important role, which is not desirable. At last, as in Fig. 2b, we could achieve similar distribution between different sensitive groups within a cluster. This can be explained by the proposed objective functions that our $\mathcal{L}_{fair}$ aims to learn the representation that follows the multivariate normal distribution for all sensitive groups meanwhile the centroids of a different sensitive group within the cluster. It is noticeable that the representation from DFC is highly identifiable compared with ours. This would result in potential bias in downstream tasks or possibly generating clusters with the same sensitive group.

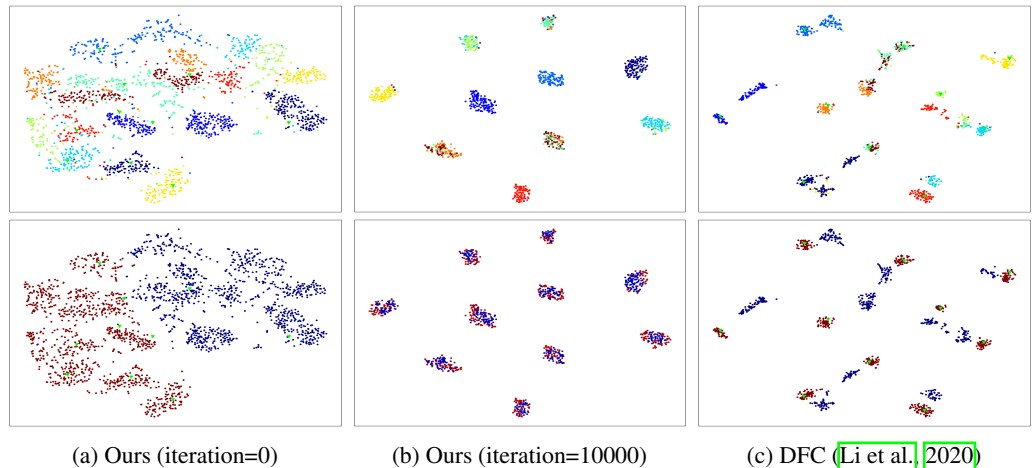

(a) Ours (iteration=0)  (b) Ours (iteration=10000)  (c) DFC (Li et al., 2020)

Figure 2: $t$-SNE visualization of FFC and DFC from 1024 randomly selected samples on MNIST-USPS dataset. Samples with different colors indicate different ground truth labels (10 digits) and different sensitive groups (MNIST, USPS) for the top and the bottom row.

| | Acc (Diff) | NMI | Balance | FFD$^2$ |
|---|---|---|---|---|
| Perfect | 1.0 (0.0) | 1.0 | 0.12 | - |
| DFC | 0.824 (0.160) | 0.828 | 0.053 | 14.13 |
| DFC ($k$-means++) | 0.812 (0.115) | 0.754 | 0.044 | 7.61 |
| Ours | 0.831 (0.015) | 0.837 | 0.091 | 1.82 |
| Ours ($k$-means++) | 0.831 (0.016) | 0.834 | 0.090 | 1.82 |

Table 3: Evaluation of the learned representations from deep networks on MNIST-USPS dataset. We compare the end-to-end deep model and adopt $k$-means++ clustering method. The representation with lower FFD achieves more stable and fair results.

## 6.4 JUSTIFICATION OF FAIR FRÉCHET DISTANCE AS A FAIRNESS METRIC

Representation learning for clustering using deep networks can benefit from their structure of discovering intrinsic features that are difficult to observe in raw data. However, as we mentioned in the motivation, if samples are computationally-identifiable (unfair), they are more vulnerable to being clustered with extrinsic features *i.e.,* sensitive attribute.

To validate this claim, we conduct a classical $k$-means++ algorithm to cluster the learned representation from our method and DFC. In Table 3, we summarize the results. As expected, DFC lost more accuracy and NMI compared to ours when the learned features are clustered with $k$-means++ because the FFD was higher than ours. In contrast, we observe almost identical results by $k$-means++ when we train with our representation. Also, we achieve better balance and NMI compared with DFC variant. This confirms that FFD is a good metric of fair clustering as the representation with lower FFD consistently outcomes fair clusters. This is also shown qualitatively by t-SNE representation. When the representation is computationally-identifiable and easily separable by the sensitive attribute, this can result in subsequent unstable and unfair clustering.

## 7 CONCLUSION AND DISCUSSION

In this paper, we define computationally unidentifiable fairness as a novel notion of fairness to measure distributional independence of sensitive attributes by leveraging Fréchet distance. Furthermore, we elaborate on the theoretical analysis of the proposed metric and find some interesting properties. We integrate contrastive learning and distributional constraint to achieve state-of-the-art performance while maintaining computational-unidentifiability. We report experimental results comparing with other fair clustering methods on various benchmark datasets to validate our claim.

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
