# OpenReview forum: "Computational-Unidentifiability in Representation for Fair Downstream Tasks"
_ICLR.cc/2023/Conference — Submitted to ICLR 2023_

### Official Review · Reviewer_7PFC · 2022-10-25

**Confidence:** 2
**Clarity, Quality, Novelty And Reproducibility:** 1. The reproducibility of the work is…
**Correctness:** 3
**Technical Novelty And Significance:** 3
**Empirical Novelty And Significance:** 3
**Recommendation:** 6

**Strength And Weaknesses:**

Strengths:
1. They propose a novel metric that quantifies fairness in representation space. And they provide rigorous analysis of the theoretical property and complexity of the fairness metric.
2. They propose a framework for fair representation learning for downstream tasks.
3. They compare the proposed method with the state-of-the-art fair method in the literature (Balance).

Weaknesses:




**Summary Of The Paper:**

This paper attempts to address the fair representation learning method for unsupervised learning. They define a notion of fairness, computational-unidentifiability, which suggests the fairness of the representation as the distributional independence of the sensitive groups. They also propose a new fairness metric, FFD, to quantify the computational-unidentifiability. Then, some comparison experiments are also conducted to validate the effectiveness of the proposed method in various downstream tasks.

**Summary Of The Review:**

This paper provides a set of solutions for fair representations, including a novel metric and a framework. Some experiments are also conducted. I think it is good enough for publication.

---

### Official Review · Reviewer_Tw9y · 2022-10-25

**Confidence:** 4
**Correctness:** 2
**Technical Novelty And Significance:** 3
**Empirical Novelty And Significance:** 4
**Recommendation:** 3

**Clarity, Quality, Novelty And Reproducibility:**

Clarity: I think the clarity of this paper could be improved greatly, particularly in terms of its proposed notion of fairness
Quality: it's hard to tell exactly the quality of the work - I think there are useful pieces in here conceptually and empirically, but don't really know how to best evaluate the whole of it
Novelty: I think there is novelty in the clustering application of a fair representation learning method, but I think the novelty is greatly overstated by the introduction of the paper
Reproducibility: some experimental details are missing, for instance, how the clustering procedure is run, and its hyperparameters

**Strength And Weaknesses:**

I think fair clustering is a useful direction and there are some novel ideas in here, but overall I'm not sure I totally grasp the direction this paper is heading in.

Some feedback:
- Introduction: the introduction makes some claims about novelty that I don't think are supported. The authors introduce "computational-unidentifiability" which they later define as the property that "no external classifier can distinguish which sensitive group the sample is drawn from" (Sec 3, Motivating Problems). They claim that this is a novel approach, and that in general, "fairness concerns in the [unsupervised learning] space have been overlooked". I think these novelty claims are misguided given the literature on adversarial fair representation learning (see Edwards & Storkey [1] for the first instance of these ideas, and Madras et al [2] for a thorough examination of an adversarial fairness metric which encapsulates this idea. There is also a literature on fair representation learning outside of adversarial notions which these novelty claims seem to ignore. Now, as far as I know, the adversarial fair representation learning literature doesn't usually consider clustering, and I believe that application is novel - however, the introduction and novelty claims of the paper should be rescoped accordingly.
- Introduction: the exposition is a little bit confusing. Re: clarity, the introduction really only mentions clustering in passing, instead discussing unsupervised learning/representation learning more generally. However, I think it would be good to focus the introduction much more clearly on clustering, given that this paper is only concerned with clustering.
- Fair Frechet Distance: I'm a little bit confused by the metric Fair Frechet Distance, and which corresponding notion of fairness it is intended to enforce. As far as I can tell, it seems like FFD is intended to minimize distributional distance between groups within a cluster. However, for the notion of a fair representation, and if we are concerned about a downstream classifier identifying these groups, shouldn't we also be concerned about the distribution of groups *between* clusters? I think there should be a little more work done in the paper to justify why this is a good metric for fairness specifically.
- Sec 5: I think reading between the lines I can figure out how the encoder and clustering objective work together, but I think it would be helpful to have some more explanation here - up to this point there is no mention of an encoder, and it would be helpful for those not so familiar with the literature to explain how it is incorporated (rather than just doing clustering on fixed representations).
- Table 2: I'm not quite sure what the takeaway is from the MTFL column - it seems that the proposed methods are much worse than a number of the baselines on FFD. There should probably be a bit more explanation about what is happening here.

Notes:
- Sec 6.1, 2nd paragraph: "desired clustering attribute is gender" - can you explain what a desired clustering attribute is here?
- some sloppiness in language throughout - for instance, bottom of page 8: "It is noticeable that the representation from DFC is highly identifiable compared with ours". I think what the authors are trying to say is that it is easier to identify, within a cluster, the sensitive attribute of interest from the DFC representations than from their proposed method's. Being a little bit more precise with expressing these notions I think will go a long way to improving the clarity of the paper
- I find section 6.4 a bit confusing - I'm not quite sure what the experimental protocol is here, and what is being compared


[1] Edwards, Harrison, and Amos Storkey. "Censoring representations with an adversary." arXiv preprint arXiv:1511.05897 (2015).
[2] Madras, David, et al. "Learning adversarially fair and transferable representations." International Conference on Machine Learning. PMLR, 2018.


**Summary Of The Paper:**

In this paper, the authors propose a method for deep clustering which attempts to match the distributions of several sensitive groups within each cluster. They propose a metric/loss based on Frechet distance, and incorporate this into a learning framework. Empirically, they show how this loss can result in clusters with the desired property.

**Summary Of The Review:**

I think this paper could use some improvement in terms of the communicating the usefulness of the fairness notion they are considering, as well as general clarity improvements around the exposition, method and experiments. As such, I'm recommending a rejection.

---

> ### Author Response · Authors · 2022-11-19
> **Response to the reviewer 1**
>
>
> **[The scope of the paper]**
>
> Thank you for the suggestion. As the reviewer mentioned, most of the fair representation learning methods have been proposed in the classification task. This motivated us to develop fair representation learning for clustering, which can be applied to _any_ downstream tasks at the time of use. Unlike other works, we do not have to constrain specific fairness metrics since the proposed method aims for fairness in the learned representation space. For example, the objective of LAFTR (Madras et al.) requires specific fairness constraints (EOd, EOp, etc) to learn fair representation adversarially. This cannot be achieved without label information. In contrast, ours enjoy the flexibility is beneficial as we can learn fair representation in _task-agnostic_ manner. We will clarify the novelty of the work lies in ``fair representation without label information'', which is not actively studied yet.
>
> **[Justification as a good metric]**
>
> Grounded on distributional independence, we propose FFD as an efficient approximation of FD to measure fairness in unsupervised learning. We claim that FFD is a good fairness metric since we can get fair results in the downstream tasks without any task-specific fairness constraint. To validate our claim, in Sec E in the appendix, we evaluated the classification and clustering result from the learned representation of DFC and ours. Since smaller FFD indicates fairer representation, regardless of the downstream task, we could achieve better fairness for each downstream task with representation with lower FFD. Specifically, in MTFL dataset, our representation has 48.64 FFD and achieves 0.129 EOD violation in the classification task, while DFC has 67.84 FFD and 0.375 EOD violation. The result suggests that representation with lower FFD (i.e., distributional independent) leads to fair downstream tasks. Therefore, we claim that FFD is a good measure of fairness to learn a fair representation.
>
> Also, note that we do not consider _the distribution of groups between clusters_.
> Our goal in this work is to achieve $P(z |A=0, C=k)=P(z |A=0, C=k)$, $\forall k$ in binary protected group scenario.
> If we want the distribution of groups between clusters to be the same as
> \begin{equation}
>     \begin{split}
>          P(z|A=0) & = \sum_k P(z |A=0, C=k) P(C=k|A=0) \\\\
>      & = \sum_k P(z |A=1, C=k) P(C=k|A=1) \\\\
>      & =  P(z|A=1),
>     \end{split}
> \end{equation}
> we would require $ P(C=k|A=0)= P(C=k|A=1)$, which is too strong an assumption since this equality may not hold in the actual data distribution.
>
> To validate the efficiency, we summarize the specification of datasets and computational time of FD and FFD in the table below. We notice that the reduction of computational time of FFD greatly improves when the dataset is more complicated (e.g., image).
> |               | MNIST-USPS           | MTFL       | Adult       |
> |---------------|-------------|-------------|-------------|
> | Sample size   |67291      | 12995     |    45211     |
> | Feature size |256      | 768     |     56     |
> |---------------|-------------|-------------|-------------|
> | Computational Time of FD        |1.25      | 1.04     |     0.065    |
> |Computational Time of  FFD           |0.15         | 0.06      |0.036    |
> |---------------|-------------|-------------|-------------|
> |Reduce rate           |88.0\%         |94.23\%     | 44.6\%   |
> |---------------|-------------|-------------|-------------|
>
>
> **[Explanation of the framework]**
>
> First, we want to clarify that the encoder in our framework follows the structure of the encoder of VAE. This deep-structured encoder allows us to leverage rich information of complicated data (e.g., image). Given the structure, our encoder outputs latent vectors as variational posterior by utilizing the reparameterization trick. This means we enforce our posteriors to follow Gaussian prior, which is parametric. It allows us to measure the distance between two latent vectors with KL divergence in a simple form as in Eqn (5), as a fair loss. With the representation $z = E(x)$ projected from the encoder, we find the centroids of each cluster as the trainable variables of the clustering module. The objective function of the clustering follows a self-training strategy, i.e., minimizing KL divergence with target distribution $q$ as sharpening of the soft assignment $p = C(z)$.
>
> **[Definition of desired clustering attribute]**
>
> The desired clustering attribute indicates the attribute that we want the representation to be clustered by. This is _digit_ for MNIST-UPSP dataset.

---

> ### Author Response · Authors · 2022-11-19
> **Reponse to the reviewer 2**
>
> **[Interpretation of Table 2]**
>
> Note that ${FFD}^2$ with underline is measured in the input space with the predicted cluster as described in the caption of Table 2. Thus, FFD from deep models and non-deep approaches cannot be compared directly since deep models use latent space and non-deep models utilize original input space. A direct comparison should be made within the same class, and ours have a lower FFD than DFC.
> Also, note that we set latent space to have same the same dimension for DFC and ours and normalized when computing FFD for a fair comparison. More details can be found in the first paragraph of Sec 6.2.
>
> **[Experimental details]**
>
> We described our experimental setup in Sec 6.1. Note that we do not require any further hyperparameter tuning except the learning rate. Since we use the same structure as DFC, we use the same learning rate for both DFC and ours: $10^{-5}$ (MTFL), $5\times 10^{-4}$ (MNIST-USPS).
>
> **[Clarity of the sentence]**
>
> Thank you for pointing this out. What the reviewer understood is correct. We will revise the draft as the following for better clarity.
> ``Comparing the learned representation from DFC with ours, it is easier to identify the sensitive group within a cluster in DFC and this indicates theirs is distributionally dependent.''
>
> **[Interpretation of Sec 6.4]**
>
> The experiment in Sec 6.4 and Sec E in the appendix is to support our claim that representation with lower FFD is fair representation in _task-agnostic_ manner.
> Here, we evaluate the learned representation in two downstream tasks. We input the learned representation for $k$-means++ and logistic regression and evaluate the performance and fairness violations.
> Specifically, in Table 3, we compare the original clustering result from DFC (second row) and ours (fourth row) with k-means++ with the learned representation from DFC (third row) and ours (fifth row). We can observe that our representation consistently achieves better fairness and performance while DFC has degradation on both fairness and performance with k-means++. This shows that higher FFD makes the downstream performance more vulnerable, as shown similarly in the toy example in Sec 3.1.

---

### Official Review · Reviewer_mwVu · 2022-10-26

**Confidence:** 3
**Correctness:** 2
**Technical Novelty And Significance:** 2
**Empirical Novelty And Significance:** 3
**Recommendation:** 3

**Clarity, Quality, Novelty And Reproducibility:**

Clarity: The paper is clearly written. I have some conceptual questions that remain unanswered

Quality: The contribution is interesting overall the paper is addressing an important problem. I do feel the number of baselines is insufficient.

Novelty: Reasonably novel, though some choices made require clarification

Reproducibility: Code is provided, I think experimental details are reasonable for reproducibility

**Strength And Weaknesses:**

1. Definition of distributional independence that is distinct from statistical independence is not clear, and seems unnecessary given that downstream task is mostly only minimizing loss terms in terms of the KL divergence.

2. The choice of KL-divergence to impose a fairness constraint is odd, and very specific. Why isn't a non-parametric distance like MMD preferred?

3. Authors don't provide any intuition on why the fair fretchet distance is a good metric beyond bounding it with the fretched distance.

4. Computational unidentifiability seems to imply that the distribution for the sensitive attributes within cluster are essentially the same. Hence I am a bit concerned about why only a limited clustering algorithm based on KL is proposed. Many distributions don't have closed form estimators, and non-parametric estimates are challenging. On the other hand, scaling MMD seems much easier.

5. More baselines could be added such as the ones authors themselves cite - fair-centroid?

6. Overall clustering being ill-defined, one could also compare all fair representation learning methods that impose direct independence between the learned representation and the outcome (since the goal seems to be to improve downstream classification fairness)



**Summary Of The Paper:**

This paper introduces computational-unidentifiability, a concept for fair representation learning which essentially makes it challenging to identify the sensitive attribute of a sample within cluster, by ensuring that the two distributions remain the same. An associated metric called the fair fretchet distance is proposed. An algorithm for clustering that minimizes the KL-divergence based fairness loss (ensuring the within cluster samples) have low KL divergence under gaussian assumptions is demonstrated to have better fair fretched distance on benchmark datasets with improvement in downstream fairness tasks

**Summary Of The Review:**

I am concerned that the contribution is lacking some conceptual nuances and choices made for the algorithm are not fully justified. Hence I will review the rebuttal to see if authors are able to address all concerns.

---

> ### Author Response · Authors · 2022-11-19
> **Response to the reviewer**
>
> **[Idea of distributional independence]**
>
> Indeed, the primary objective is usually minimizing KL-divergence (e.g., cross-entropy in classification and Eqn. (4) in clustering as in the main paper) between target distribution and model output. However, we need to meet the parity between some quantitative measures to achieve group fairness. For instance, demographic parity in classification is to achieve predictive parity, i.e., $P(\hat{Y}=1|A=1)=P(\hat{Y}=1|A=0)$, and balance is to achieve a similar objective in clustering, i.e., $P(A=1|\hat{C}=c)=P(A=0|\hat{C}=c)$, where $\hat{Y}$ and $\hat{C}$ are predicted class and cluster, respectively. However, what we claim is that such statistical parity has some limitations. For example, suppose we can get 0 KL-divergence at inference (perfect model); we cannot achieve perfect demographic parity or balance if _base rate_ between different protected groups is different. Thus, we propose to achieve _distributional independence_ instead of _statistical parity_ to quantify fairness violation. Similarly, FairGAN [1] proposed to generate images from different protected groups to follow the same distribution so that the generative model can generate fair data.
>
> **[Justification of using KL divergence]**
>
> First, we want to clarify that the encoder in our framework follows the structure of the encoder of VAE. Given the structure, our encoder outputs latent vectors as variational posterior by utilizing the reparameterization trick. Thus we enforce our posteriors to follow Gaussian prior, which is parametric. The prior regularization allows us to measure the distance between two latent vectors with KL divergence in a simple form as in Eqn (5). MMD can be an alternative; however, we can accurately compute KL divergence more efficiently than MMD in this case.
>
> **[Justification of why FFD is a good fair metric]**
>
> Grounded on distributional independence, we propose FFD as an efficient approximation of FD to measure fairness in unsupervised learning. We claim that FFD is a good fairness metric since we can get fair results in the downstream tasks without any task-specific fairness constraint. To validate our claim, in Sec E in the appendix, we evaluated the classification and clustering result from the learned representation of DFC and ours. Since smaller FFD indicates fairer representation, regardless of the downstream task, we could achieve better fairness for each downstream task with representation with lower FFD. Specifically, in MTFL dataset, our representation has 48.64 FFD and achieves 0.129 EOD violation in the classification task, while DFC has 67.84 FFD and 0.375 EOD violation. The result suggests that representation with lower FFD (i.e., distributional independent) leads to fair downstream tasks. Therefore, we claim that FFD is a good measure of fairness to learn a fair representation.
>
> To validate the efficiency, we summarize the specification of datasets and computational time of FD and FFD in the table below. We notice that the reduction of computational time of FFD dramatically improves when the dataset is more complicated (e.g., image).
> || MNIST-USPS| MTFL| Adult|
> |-|-|-|-|
> | Sample size |67291| 12995|45211|
> | Feature size |256| 768| 56 |
> | Computational Time of FD |1.25 | 1.04| 0.065|
> |Computational Time of FFD|0.15| 0.06| 0.036|
> |Reduce rate |88.0\% |94.23\%| 44.6\%|
>
>
> **[Difference with other fair representation learning methods]**
>
> We want to clarify that our objective is to learn fair representation without the label information. Previously, most of the fair representation learning methods are proposed in the classification task. This motivated us to develop fair representation learning for clustering, which can be applied to _any_ downstream tasks at the time of use. Unlike other works, we do not have to constrain specific fairness metrics since the proposed method aims for fairness in the learned representation space. This flexibility is beneficial as we can learn fair representation in _task-agnostic_ manner. This claim is supported by the toy example in Sec 3.1 in the main paper.
>
>
> **[Additional experiments]**
>
> Thank you for the suggestion. We included the result of fair-centroid as follows.
> We observe that the proposed method outperforms both accuracy and fairness.
> | MNIST-USPS| Acc| NMI| Balance | $FFD^2$|
> |-|-|-|-|-|
> | Fair-centroid [2] |0.725    | 0.797     |     0.071     |   1.99 |
> | DFC |0.824 | 0.828|0.053    | 14.13 |
> | Ours|0.831| 0.837     | 0.091   |1.82    |
>
>
> | MTFL | Acc  | NMI | Balance | $FFD^2$|
> |-|-|-|-|-|
> | Fair-centroid [2] |0.725    |0.149| 0.101   | 44.01 |
> | DFC     |0.731| 0.163|0.137 | 48.02 |
> | Ours    |0.728 | 0.154 | 0.113 |43.58|
>
> [1] Xu, Depeng et al. Fairgan: Fairness-aware generative adversarial networks, IEEE International Conference on Big Data (Big Data), 2018
>
> [2] Bokun Wang and Ian Davidson. Towards fair deep clustering with multi-state protected variables, arXiv preprint arXiv:1901.10053, 2019

---

### Decision · Program_Chairs · 2023-01-20

**Decision:**

Reject

**Justification For Why Not Higher Score:**

There was no strong support for accepting this paper and several concerns among reviewers.

**Justification For Why Not Lower Score:**

N/A

**Metareview: Summary, Strengths And Weaknesses:**

This paper proposes a notion and metric for fair representations. While there were some strengths to the paper, there was no strong support for accepting it. Instead, reviewers brought up several concerns: 1) Overstated novelty in the introduction. 2) The clarity of the paper is not yet at the level expected for publication. 3) Several choices in the algorithm are insufficiently justified. 4) The paper has insufficient comparisons with baselines. As such, I recommend rejecting this paper.